# Arrival Time from Hamiltonian with Non-Hermitian Boundary Term

Tajron Jurić * and Hrvoje Nikolić

Theoretical Physics Division, Rudjer Bošković Institute, P.O. Box 180, HR-10002 Zagreb, Croatia; hnikolic@irb.hr
* Correspondence: tjuric@irb.hr

**Abstract:** In this study, we developed a new method for finding the quantum probability density of arrival at the detector. The evolution of the quantum state restricted to the region outside of the detector is described by a restricted Hamiltonian that contains a non-Hermitian boundary term. The non-Hermitian term is shown to be proportional to the flux of the probability current operator through the boundary, which implies that the arrival probability density is equal to the flux of the probability current.

**Keywords:** arrival time; non-Hermitian Hamiltonian; boundary term; probability current

## 1. Introduction

Consider a quantum particle described by a spatially extended wave packet impinging on the detector region $D$. Since different parts of the packet approach $D$ at different times, there is an inherent quantum uncertainty about the time at which the particle arrival to $D$ will be detected. The arrival time problem refers to making a theoretical prediction for the probability distribution $\mathcal{P}_{\mathrm{arr}}(t)$ that the arrival will be detected at time $t$. Remarkably, there are many different theoretical approaches to this problem, which make different measurable predictions (for reviews see [1–4]), and it is not clear, neither theoretically nor experimentally, which approach is correct.

In general, quantum mechanics makes unambiguous probabilistic predictions for various phenomena; so why is the arrival time problem a problem at all? The central point is that quantum mechanics makes unambiguous probabilistic predictions for measurements of observables represented by self-adjoint operators, while time, in the usual formulation of quantum mechanics, is not an observable in this sense. Time is a classical parameter, not a quantum operator; so, by starting from general axioms of quantum theory, it is not immediately clear how to make quantum probabilistic predictions associated with the measurement of time. In particular, one does not know at which time a quantum event, such as a particle detection, will happen, so one must use quantum mechanics to compute a probability that the event will happen at a given time. The problem then is how to compute this probability when the time is not an operator. The arrival time problem is the simplest version of this problem, where the quantum event is taken to be the particle arrival to the detector, or more operationally, a click in the detector, which happens when the particle arrives.

One class of possibilities (see, e.g., [1,4] and the references therein) is to reformulate quantum mechanics such that time is treated as an operator. However, the problem with such approaches is that they may require a radical reformulation of the general principles of quantum mechanics, which makes them rather controversial. There are also axiomatic approaches, such as those by Kijowski and others (see, e.g., [1,4] and the references therein) that postulate axioms for the arrival time distribution. However, the problem is that these axioms seem somewhat ad hoc because they cannot be derived from the standard axioms of quantum mechanics. Another class of possibilities (see, e.g., [4] and the references therein)

is that of semi-classical approaches. However, the problem here is that they also seem too ad hoc and lack a deeper understanding of the problem.

Yet another class [5–12] of approaches to the arrival time problem predicts that $\mathcal{P}_{\mathrm{arr}}(t)$ is given by the flux of the probability current. Within this class, some approaches are based on standard quantum mechanics (QM) [5–8], while others are based on the Bohmian formulation of QM in terms of particle trajectories [9–12]. In this paper, we present one new approach to the arrival time problem, based on standard QM, which confirms that the arrival time distribution is given by the flux of the probability current. Given that it is not generally accepted in the community that the arrival time distribution should be given by the flux of the probability current, we believe that it is valuable to present one more piece of independent theoretical evidence that it is indeed so.

The approach in this paper was partially inspired by the approach in [8], but was motivated with the goal of avoiding certain mathematical subtleties that appeared in that work. The approach in [8], which arose from the development of earlier ideas in [13,14], is based on time evolution governed by a projected Hamiltonian $\overline{H} = \bar{\pi} H \bar{\pi}$, where $\bar{\pi}$ is the projector to the region $\bar{D}$ defined as the complement of the detector region $D$. The mathematical subtleties appear because, in the position representation, $\bar{\pi}$ is represented by a characteristic function with a discontinuity at the boundary of $\bar{D}$, which leads to ambiguities when the second-derivative operator appearing in $H$ acts on a function with a discontinuity. The goal of this paper was to develop a formalism based on an alternative definition of $\overline{H}$ that, at the same time, captures the same physics as $\overline{H}$ in [8,13,14], but uses a different mathematical definition of $\overline{H}$ so that the mathematical difficulties appearing in [8,14] are avoided.

The paper is organized as follows. In Section 2, we start by defining the notion of the restricted wave function, which in $\bar{D}$ coincides with the full wave function, but vanishes outside of $\bar{D}$ where the full wave function, in general, does not vanish. This implies that the norm of the restricted wave function, in general, is not conserved in time. Then, in Section 3, we define the restricted Hamiltonian $\overline{H}$, which governs the evolution of the restricted wave function. It turns out that the restricted Hamiltonian is non-trivial at the boundary of $\bar{D}$, because it has a non-Hermitian boundary term proportional to the flux of the probability current operator, which accounts for the non-conservation of the norm of the restricted wave function. In Section 4, we explain how this non-conservation of the norm implies that the arrival time distribution is equal to the flux of the probability current. The conclusions are drawn in Section 5, and, in Appendix A, an alternative derivation of the non-Hermitian part is presented.

## 2. Restricted Wave Function

Let us start with an elementary review to establish the notation. Consider a particle moving in a 3-dimensional space $\mathbb{R}^3$. It is described by a wave function

$$\psi(\mathbf{x}, t) = \langle \mathbf{x} | \psi(t) \rangle, \tag{1}$$

where $|\psi(t)\rangle$ is the state in the Hilbert space $\mathcal{H}$ denoted in Dirac's "bra-ket" formalism. Here, $|\psi(t)\rangle$ and $\psi(\mathbf{x}, t)$ satisfy the respective Schrödinger equations

$$H|\psi(t)\rangle = i\partial_t |\psi(t)\rangle, \quad \hat{H}\psi(\mathbf{x}, t) = i\partial_t \psi(\mathbf{x}, t), \tag{2}$$

where $H$ is an abstract operator, while $\hat{H}$ is its coordinate representation given by a concrete derivative operator

$$H = \frac{\mathbf{p}^2}{2m} + V(\mathbf{x}), \quad \hat{H} = -\frac{\boldsymbol{\nabla}^2}{2m} + V(\mathbf{x}), \tag{3}$$

and we work in units: $\hbar = 1$. The relation between $H$ and $\hat{H}$ can be expressed as

$$\langle \mathbf{x} | H | \psi(t) \rangle = \hat{H}\psi(\mathbf{x}, t). \tag{4}$$

Now, after this review, let us divide the full space $\mathbb{R}^3$ into a detector region $D$ and its complement $\bar{D}$, so that $D \cup \bar{D} = \mathbb{R}^3$, $D \cap \bar{D} = \emptyset$. Physically, such a division is motivated with the goal of studying the arrival of the particle to the detector. We define the *restricted wave function* $\psi_{\bar{D}}(\mathbf{x}, t)$ as

$$\psi_{\bar{D}}(\mathbf{x}, t) \equiv \begin{cases} \psi(\mathbf{x}, t) & \text{for } \mathbf{x} \in \bar{D} \\ 0 & \text{for } \mathbf{x} \notin \bar{D}. \end{cases} \tag{5}$$

It can be expressed as

$$\psi_{\bar{D}}(\mathbf{x}, t) = \langle \mathbf{x} | \bar{\pi} | \psi(t) \rangle, \tag{6}$$

where

$$\bar{\pi} = \int_{\bar{D}} d^3 x \, |\mathbf{x}\rangle\langle\mathbf{x}| \tag{7}$$

is the projector to $\bar{D}$. Thus, we see that the restricted wave function can also be expressed as

$$\psi_{\bar{D}}(\mathbf{x}, t) = \langle \mathbf{x} | \psi_{\bar{D}}(t) \rangle, \tag{8}$$

where

$$|\psi_{\bar{D}}(t)\rangle = \bar{\pi} |\psi(t)\rangle. \tag{9}$$

Since $\psi_{\bar{D}}(\mathbf{x}, t)$ coincides with $\psi(\mathbf{x}, t)$ in $\bar{D}$, but vanishes outside of $\bar{D}$, we have

$$\int_{\mathbb{R}^3} d^3 x \, |\psi_{\bar{D}}(\mathbf{x}, t)|^2 \leq \int_{\mathbb{R}^3} d^3 x \, |\psi(\mathbf{x}, t)|^2. \tag{10}$$

In fact, since we assume that $\psi(\mathbf{x}, t)$ is a traveling wave packet (rather than a stationary state), one expects that the left-hand side depends on time $t$, despite the fact that the right-hand side is time independent. This means that the norm of $\psi_{\bar{D}}(\mathbf{x}, t)$ is expected to change with time, i.e., that the norm $\|\psi_{\bar{D}}(t)\| = \langle \psi_{\bar{D}}(t) | \psi_{\bar{D}}(t) \rangle^{1/2}$ is time-dependent. Consequently, one expects that the evolution of $|\psi_{\bar{D}}(t)\rangle$ is not unitary. By contrast, the evolution of the full state $|\psi(t)\rangle$ is unitary.

The restricted wave function can also be interpreted in terms of wave function collapse. When a part of the wave function enters the detector region $D$, there is a non-zero probability that the detector will detect the particle, i.e., that the wave function will collapse to the region $D$. However, there is also a probability that the detector will not detect the particle; in which case, we know that the particle is still outside of the detector region $D$, so the wave function collapses to the region $\bar{D}$. Thus, the restricted wave function can be interpreted as the collapsed wave function, corresponding to a negative measurement outcome by the detector. For more details of this interpretation, see [8].

From a theoretical point of view, the exact specification of the detector region $D$ in our analysis remains somewhat ambiguous. In principle, one could define it using a more detailed model. But in practice, we believe that an experimental approach would be more fruitful. One could take an actual detector and impinge on it particles with wave functions that are very narrow in the position space, so that the particles are effectively "classical" in the sense that their arrival time can be predicted as a classical deterministic event. In this way, one can determine the relevant detector region $D$ experimentally for the specific detector at hand. After that, once $D$ is known, one can conduct a non-trivial theoretical analysis with "truly quantum" non-narrow wave functions.

## 3. Restricted Hamiltonian

The full state evolves with time as $|\psi(t)\rangle = e^{-iHt}|\psi(0)\rangle$. This evolution is unitary because the Hamiltonian $H$ is Hermitian. On the other hand, since one expects that the evolution of the restricted state $|\psi_{\bar{D}}(t)\rangle$ is not unitary, its evolution can be described as

$$|\psi_{\bar{D}}(t)\rangle = e^{-i\bar{H}t}|\psi_{\bar{D}}(0)\rangle, \tag{11}$$

where $\overline{H}$ is expected to be some *non-Hermitian* operator. Our goal was to find an explicit expression for $\overline{H}$.

Heuristically, since $\psi_{\bar{D}}(\mathbf{x}, t)$ and $\psi(\mathbf{x}, t)$ coincide for $\mathbf{x} \in \bar{D}$, the coordinate representation $\hat{\overline{H}}$ must coincide with the derivative operator $\hat{H}$ in (3) for $\mathbf{x} \in \bar{D}$. Similarly, since $\psi_{\bar{D}}(\mathbf{x}, t) = 0$ for $\mathbf{x} \notin \bar{D}$, the operator $\hat{\overline{H}}$ can be taken as the trivial zero operator for $\mathbf{x} \notin \bar{D}$. However, particular care should be taken concerning the definition of $\hat{\overline{H}}$ at the *boundary* of $\bar{D}$, which is the only place where subtleties in the definition of $\hat{\overline{H}}$ can be expected. For this purpose, we found it more convenient to work with the abstract $\overline{H}$ operator, rather than its coordinate representation $\hat{\overline{H}}$. Thus, since the arbitrary matrix element of $H$ is

$$\langle \psi_b | H | \psi_a \rangle = \int_{\mathbb{R}^3} d^3 x \, \psi_b^*(\mathbf{x}) \left[ -\frac{\boldsymbol{\nabla}^2}{2m} + V(\mathbf{x}) \right] \psi_a(\mathbf{x}), \tag{12}$$

we postulate that the arbitrary matrix element of $\overline{H}$ is

$$\langle \psi_b | \overline{H} | \psi_a \rangle \equiv \int_{\bar{D}} d^3 x \, \psi_b^*(\mathbf{x}) \left[ -\frac{\boldsymbol{\nabla}^2}{2m} + V(\mathbf{x}) \right] \psi_a(\mathbf{x}), \tag{13}$$

which has the same form as (12), except that the integration region is restricted from $\mathbb{R}^3$ to $\bar{D}$. Hence, we refer to $\overline{H}$ as the *restricted Hamiltonian*. The goal now is to find the explicit operator representation of $\overline{H}$, which is analogous to $H$ in (3).

We first write (13) as

$$\langle \psi_b | \overline{H} | \psi_a \rangle = -\frac{1}{2m} \int_{\bar{D}} d^3 x \, \psi_b^* \boldsymbol{\nabla}^2 \psi_a + \overline{V}_{ba}, \tag{14}$$

where

$$\overline{V}_{ba} = \int_{\bar{D}} d^3 x \, \psi_b^* V \psi_a = \int_{\bar{D}} d^3 x \, V \psi_b^* \psi_a = \int_{\bar{D}} d^3 x \, \psi_b^* \psi_a V. \tag{15}$$

Hence, partial integration and the Gauss theorem give

$$
\begin{aligned}
\langle \psi_b | \overline{H} | \psi_a \rangle &= \frac{1}{2m} \int_{\bar{D}} d^3 x \, (\boldsymbol{\nabla} \psi_b)^* (\boldsymbol{\nabla} \psi_a) \\
&\quad - \frac{1}{2m} \int_{\bar{D}} d^3 x \, \boldsymbol{\nabla} (\psi_b^* \boldsymbol{\nabla} \psi_a) + \overline{V}_{ba} \\
&= \frac{1}{2m} \int_{\bar{D}} d^3 x \, (\boldsymbol{\nabla} \psi_b)^* (\boldsymbol{\nabla} \psi_a) \\
&\quad - \frac{1}{2m} \int_{\partial \bar{D}} d\mathbf{S} \cdot (\psi_b^* \boldsymbol{\nabla} \psi_a) + \overline{V}_{ba},
\end{aligned}
\tag{16}
$$

where $\partial \bar{D}$ is the boundary of $\bar{D}$ and $d\mathbf{S}$ is the area element directed outwards from $\bar{D}$. Next, we use the identities

$$
\begin{aligned}
\psi_a(\mathbf{x}) = \langle \mathbf{x} | \psi_a \rangle, \quad \psi_b^*(\mathbf{x}) = \langle \psi_b | \mathbf{x} \rangle, \\
-i \boldsymbol{\nabla} \psi_a(\mathbf{x}) = \langle \mathbf{x} | \mathbf{p} | \psi_a \rangle, \quad i \boldsymbol{\nabla} \psi_b^*(\mathbf{x}) = \langle \psi_b | \mathbf{p} | \mathbf{x} \rangle,
\end{aligned}
\tag{17}
$$

implying that (16) can be written as

$$
\begin{aligned}
\langle \psi_b | \overline{H} | \psi_a \rangle &= \frac{1}{2m} \langle \psi_b | \mathbf{p} \bar{\pi} \mathbf{p} | \psi_a \rangle \\
&\quad - \frac{i}{2m} \int_{\partial \bar{D}} d\mathbf{S} \cdot \langle \psi_b | \mathbf{x} \rangle \langle \mathbf{x} | \mathbf{p} | \psi_a \rangle \\
&\quad + \langle \psi_b | \bar{\pi} V | \psi_a \rangle.
\end{aligned}
\tag{18}
$$

Since $\bar{\pi}$ commutes with $V$, the last term in (18) can also be written in alternative forms:

$$\langle \psi_b | \bar{\pi} V | \psi_a \rangle = \langle \psi_b | V \bar{\pi} | \psi_a \rangle = \langle \psi_b | \bar{\pi} V \bar{\pi} | \psi_a \rangle, \tag{19}$$

thus, we see that $\overline{H}$ can be written in the operator form

$$\overline{H} = \frac{\mathbf{p} \bar{\pi} \mathbf{p}}{2m} - \frac{i}{2m} \int_{\partial \bar{D}} d\mathbf{S} \cdot |\mathbf{x}\rangle \langle \mathbf{x} | \mathbf{p} + \bar{\pi} V \bar{\pi}. \tag{20}$$

The Hermitian conjugation gives

$$\overline{H}^{\dagger} = \frac{\mathbf{p} \bar{\pi} \mathbf{p}}{2m} + \frac{i}{2m} \int_{\partial \bar{D}} d\mathbf{S} \cdot \mathbf{p} |\mathbf{x}\rangle \langle \mathbf{x} | + \bar{\pi} V \bar{\pi}, \tag{21}$$

thus, we see that the first and the last term are Hermitian operators, but that the middle term is not. This shows that the restricted Hamiltonian $\overline{H}$ is not a Hermitian operator, owing to the boundary term.

To better isolate the source of non-Hermiticity, it is useful to write $\overline{H}$ as

$$\overline{H} = \frac{\overline{H} + \overline{H}^{\dagger}}{2} + \frac{\overline{H} - \overline{H}^{\dagger}}{2}, \tag{22}$$

which is convenient because the first term is manifestly Hermitian and the second term manifestly anti-Hermitian. From (20) and (21), we see that the two terms in (22) can be written as

$$\frac{\overline{H} + \overline{H}^{\dagger}}{2} = \frac{\mathbf{p} \bar{\pi} \mathbf{p}}{2m} + \frac{1}{2} \int_{\partial \bar{D}} d\mathbf{S} \cdot \mathbf{K} + \bar{\pi} V \bar{\pi},$$

$$\frac{\overline{H} - \overline{H}^{\dagger}}{2} = -\frac{i}{2} \int_{\partial \bar{D}} d\mathbf{S} \cdot \mathbf{J}, \tag{23}$$

where

$$\mathbf{K}(\mathbf{x}) = \frac{i[\mathbf{p}, |\mathbf{x}\rangle \langle \mathbf{x}|]}{2m}, \quad \mathbf{J}(\mathbf{x}) = \frac{\{\mathbf{p}, |\mathbf{x}\rangle \langle \mathbf{x}|\}}{2m} \tag{24}$$

are Hermitian operators, $[A, B] = AB - BA$ denotes a commutator, and $\{A, B\} = AB + BA$ denotes an anti-commutator. Thus, (22) can be written in the final form

$$\overline{H} = \left[ \frac{\mathbf{p} \bar{\pi} \mathbf{p}}{2m} + \bar{\pi} V \bar{\pi} \right] + \frac{1}{2} \int_{\partial \bar{D}} d\mathbf{S} \cdot \mathbf{K} - \frac{i}{2} \int_{\partial \bar{D}} d\mathbf{S} \cdot \mathbf{J}. \tag{25}$$

The first term (namely, the term in square brackets) is Hermitian and does not depend on the boundary. It is non-negative, provided that $V$ is non-negative. The second term (namely, the term involving $\mathbf{K}$) is a Hermitian, but not non-negative, boundary term. The last term (namely, the term involving $\mathbf{J}$) is an anti-Hermitian boundary term.

Equation (25) represents the main novelty of this paper, so let us discuss its significance qualitatively. While the full Hamiltonian (3) describes the evolution of the full wave function everywhere in full 3-dimensional space, (25) is the restricted Hamiltonian, which describes the evolution of the restricted wave function, namely, the part of the wave function defined only on the 3-dimensional region $\bar{D}$. The $\bar{\pi}$ is the projector to the region $\bar{D}$, so the term in square brackets in (25) is just the projected version of (3). The projector $\bar{\pi}$ commutes with $V = V(\mathbf{x})$, but does not commute with $\mathbf{p}$. Hence, the potential energy term $\bar{\pi} V \bar{\pi}$ can also be written as $\bar{\pi} V$ or $V \bar{\pi}$, but the kinetic energy term proportional to $\mathbf{p} \bar{\pi} \mathbf{p}$ must we written in that form, and not, e.g., as $\bar{\pi} \mathbf{p} \bar{\pi} \mathbf{p} \bar{\pi}$ or $\mathbf{p} \bar{\pi} \mathbf{p} \bar{\pi}$. The commutator $[\bar{\pi}, \mathbf{p}]$, in the $\mathbf{x}$-representation, is proportional to a Dirac $\delta$-function on the boundary of $\bar{D}$, so replacing $\mathbf{p} \bar{\pi} \mathbf{p}$ with $\bar{\pi} \mathbf{p} \bar{\pi} \mathbf{p} \bar{\pi}$ or $\mathbf{p} \bar{\pi} \mathbf{p} \bar{\pi}$ would produce spurious boundary terms. In (25), all boundary terms are represented explicitly and unambiguously, without $\delta$-functions, as surface integrals over the boundary $\partial \bar{D}$ of $\bar{D}$. The most important feature of the boundary term is the fact that it contains an

anti-Hermitian part involving **J**, the physical significance of which we discuss in more detail in the subsequent sections.

Note that the mean value of **J** is (see also [15])

$$
\begin{aligned}
\mathbf{j}(\mathbf{x}) &\equiv \langle\psi|\mathbf{J}(\mathbf{x})|\psi\rangle \\
&= \frac{-i}{2m}[\psi^*(\mathbf{x})\boldsymbol{\nabla}\psi(\mathbf{x}) - (\boldsymbol{\nabla}\psi^*(\mathbf{x}))\psi(\mathbf{x})],
\end{aligned}
\tag{26}
$$

which is the standard probability current in quantum mechanics. For that reason, we refer to **J** as the *probability current operator*.

As we said, the fact that $\overline{H}$ in (11) is not Hermitian implies that the norm

$$
\langle\psi_{\bar{D}}(t)|\psi_{\bar{D}}(t)\rangle = \langle\psi_{\bar{D}}(0)|e^{i\overline{H}^\dagger t}e^{-i\overline{H}t}|\psi_{\bar{D}}(0)\rangle
\tag{27}
$$

is not conserved in time. Since [1]

$$
\begin{aligned}
\partial_t\left(e^{i\overline{H}^\dagger t}e^{-i\overline{H}t}\right) &= \left(\partial_t e^{i\overline{H}^\dagger t}\right)e^{-i\overline{H}t} + e^{i\overline{H}^\dagger t}\left(\partial_t e^{-i\overline{H}t}\right) \\
&= e^{i\overline{H}^\dagger t}i\overline{H}^\dagger e^{-i\overline{H}t} - e^{i\overline{H}^\dagger t}i\overline{H}e^{-i\overline{H}t} \\
&= e^{i\overline{H}^\dagger t}i\left(\overline{H}^\dagger - \overline{H}\right)e^{-i\overline{H}t} \\
&= e^{i\overline{H}^\dagger t}\left(-\int_{\partial\bar{D}}d\mathbf{S}\cdot\mathbf{J}\right)e^{-i\overline{H}t},
\end{aligned}
\tag{28}
$$

where, in the last equality, we used (23), we see that (27) implies

$$
\begin{aligned}
\frac{d}{dt}\langle\psi_{\bar{D}}(t)|\psi_{\bar{D}}(t)\rangle &= -\langle\psi_{\bar{D}}(t)|\int_{\partial\bar{D}}d\mathbf{S}\cdot\mathbf{J}(\mathbf{x})|\psi_{\bar{D}}(t)\rangle \\
&= -\int_{\partial\bar{D}}d\mathbf{S}\cdot\langle\psi_{\bar{D}}(t)|\mathbf{J}(\mathbf{x})|\psi_{\bar{D}}(t)\rangle \\
&= -\int_{\partial\bar{D}}d\mathbf{S}\cdot\mathbf{j}_{\bar{D}}(\mathbf{x},t).
\end{aligned}
\tag{29}
$$

Note that the wave functions in (16) are full wave functions, not restricted wave functions. The full wave functions are assumed to be twice differentiable at the boundary of $\bar{D}$. This is because the volume integral can only be turned into the surface integral via the Gauss theorem in this case . On the other hand, the current $\mathbf{j}_{\bar{D}}$ in (29) is expressed in terms of the restricted wave function $\psi_{\bar{D}}$, which has a discontinuity at $\bar{D}$, implying that it is not differentiable. To avoid this apparent inconsistency, we must be more careful in specifying what we mean by integral over the "boundary". This really means that the surface of integration $\partial\bar{D}$ is located *infinitesimally away* from the boundary towards the interior of $\bar{D}$, where $\psi_{\bar{D}}$ coincides with $\psi$. The consequence is that $\mathbf{j}_{\bar{D}}$ in (29) coincides with $\mathbf{j}$ defined by (26), implying that (29) can finally be written as

$$
\frac{d}{dt}\langle\psi_{\bar{D}}(t)|\psi_{\bar{D}}(t)\rangle = -\int_{\partial\bar{D}}d\mathbf{S}\cdot\mathbf{j}(\mathbf{x},t).
\tag{30}
$$

This shows that the rate of change of the norm of the state restricted to the region $\bar{D}$ is given by the flux of the probability current through the boundary of $\bar{D}$.

Physically, the most important consequence of the evolution governed by the restricted Hamiltonian (25) with an anti-Hermitian boundary term is the change of the norm of the restricted wave function, as described by (30). The result (30) is rather intuitive, it can be visualized as a wave function leaking from the region $\bar{D}$, where the flux of the probability current quantifies how much of the wave function leaks through the boundary of $\bar{D}$.

### 4. Arrival Time Distribution

Suppose that at the initial time $t = 0$, the particle is outside of the detector region $D$. This means that

$$|\psi(0)\rangle = |\psi_{\bar{D}}(0)\rangle, \tag{31}$$

i.e., the initial full state is equal to the initial state restricted to $\bar{D}$. Then, (27) is the probability $\bar{P}(t)$ that, at time $t$, the particle is in $\bar{D}$

$$\bar{P}(t) = \langle\psi_{\bar{D}}(t)|\psi_{\bar{D}}(t)\rangle. \tag{32}$$

Hence, the probability that the particle is in the detector region $D$ is

$$P(t) = 1 - \bar{P}(t). \tag{33}$$

Now, suppose that, during a time interval $[0, T]$, the probability $P(t)$ increases with time. Then, there is a positive function $\mathcal{P}(t)$ such that

$$P(t) = \int_0^t dt' \, \mathcal{P}(t'), \tag{34}$$

for any $t \in [0, T]$. This, together with (33), implies

$$\mathcal{P}(t) = \frac{dP(t)}{dt} = -\frac{d\bar{P}(t)}{dt}. \tag{35}$$

Using (32) and (30), this finally gives

$$\mathcal{P}(t) = \int_{\partial\bar{D}} d\mathbf{S} \cdot \mathbf{j}(\mathbf{x}, t). \tag{36}$$

Mathematically, the final formula (36) is rather compact and general. The same formula was also obtained in [8] using different methods, while here we derived it through the use of the restricted Hamiltonian (25) with the anti-Hermitian boundary term.

Since $P(t)$ is a probability, it follows that $\mathcal{P}(t)$ in (34) is a probability *density*. In other words, $\mathcal{P}(t)$ is a probability distribution. The question is: a probability distribution of *what*? We shall present two independent arguments: one heuristic and the other more rigorous, that $\mathcal{P}(t)$ is the probability distribution of arrival times to the detector.

For the heuristic argument, consider first an analogous quantum equation for spatial distributions. In a formula of the form $P = \int d^3x\, |\psi(\mathbf{x})|^2$, the quantity $|\psi(\mathbf{x})|^2$ is the probability density that the particle will *appear* at the position $\mathbf{x}$, rather than at any other position $\mathbf{x}'$. By analogy, $\mathcal{P}(t)$ in (34) is the probability density that the particle will *appear* at the time $t$, rather than at any other time $t'$. More precisely, since $P(t)$ is the probability that the particle is in the detector region $D$, it follows that $\mathcal{P}(t)$ is the probability density that the particle will *appear* at time $t$ in the detector region $D$. The appearance of a particle in the detector at time $t$ means that the particle was not there immediately before $t$, so we can say that the particle *arrives* to detector at time $t$. Hence, we conclude that (36) is the *arrival time distribution*.

We repeat that this interpretation is only valid when $P(t)$ increases with time, i.e., when $\mathcal{P}(t)$ is positive. Formula (36) then says that the arrival time distribution is given by the flux of the probability current through the boundary of the detector, when the flux is positive. But what if the flux is negative? In that case, $P(t)$ *decreases* with time, rather than increases, so the particle departs from the detector, rather than arrives to it. Hence, for a negative flux, the arrival probability density is zero. In this case, the $-\int_{\partial\bar{D}} d\mathbf{S} \cdot \mathbf{j}(\mathbf{x}, t)$ is positive and naturally interpreted as the departure probability density [8].

Now, let us confirm the conclusion above, i.e., that $\mathcal{P}(t)$ is arrival probability density, using a more rigorous analysis. We first split the time interval $[0, t]$ into $k$ intervals, each of the small size $\delta t = t/k$, and imagine that the particle can only arrive at one of the times

from the discrete set $t_1 = \delta t$, $t_2 = 2\delta t, \ldots, t_k = k\delta t = t$. At the end, we shall let $\delta t \to 0$. Let $w(t_j)$ be the *conditional* probability density that the particle is in the detector at time $t_j$, *given* that it was not in the detector immediately before, at time $t_{j-1}$. Then, the probability that it will arrive at time $t = t_k$ is

$$\mathcal{P}_{\mathrm{arr}}(t)\delta t = w(t)\delta t\, \bar{P}(t - \delta t), \tag{37}$$

where $\bar{P}(t - \delta t)$ is the probability that, at time $t - \delta t$, the particle was not in the detector region $D$ (see (33)). But the probability $\bar{P}(t - \delta t)$ is itself a joint probability that the particle was not in $D$ at time $t - \delta t$ *given* that it was not there at $t - 2\delta t$, *and* that it was not there at $t - 2\delta t$ *given* that it was not there at $t - 3\delta t$, etc. Hence,

$$\bar{P}(t - \delta t) = \prod_{j=1}^{k-1}[1 - w(t_j)\delta t], \tag{38}$$

where $1 - w(t_j)\delta t$ is the conditional probability that the particle is not in $D$ at time $t_j$, given that it was not there at $t_{j-1}$. Since we are interested in the limit $\delta t \to 0$, we can first write (38) as

$$\begin{aligned}
\bar{P}(t - \delta t) &= \prod_{j=1}^{k-1}\exp\left(-w(t_j)\delta t + \mathcal{O}(\delta t^2)\right) \\
&= \exp\left(-\sum_{j=1}^{k-1}[w(t_j)\delta t + \mathcal{O}(\delta t^2)]\right),
\end{aligned} \tag{39}$$

and then take the limit $\delta t \to 0$, which gives

$$\bar{P}(t - \delta t) = \bar{P}(t) = e^{-\int_0^t dt'\, w(t')}. \tag{40}$$

Thus, (37) in the limit $\delta t \to 0$ can be written as

$$\begin{aligned}
\mathcal{P}_{\mathrm{arr}}(t) &= w(t)\bar{P}(t) = w(t)e^{-\int_0^t dt'\, w(t')} \\
&= -\frac{d\bar{P}(t)}{dt} = \mathcal{P}(t),
\end{aligned} \tag{41}$$

where, in the last equality, we used (35). This shows that (36) is indeed the arrival probability density, provided that it is positive.

The measurable predictions of the arrival time distribution based on the flux of the probability current can, in principle, be distinguished experimentally from predictions of the arrival time distribution based on other approaches. It is beyond the scope of the present paper to discuss such measurable differences in detail, but they have been studied elsewhere [4].

## 5. Summary and Conclusions

The results of this paper can be summarized as follows. As the wave function of a particle approaches the detector, a part of the full wave function leaks into the detector region $D$, so the other part of wave function, which remains outside of $D$, diminishes with time. Since the norm of the full wave function $\psi(\mathbf{x}, t)$ does not depend on time, the norm of its restriction $\psi_{\bar{D}}(\mathbf{x}, t)$ to the region $\bar{D}$ outside of the detector depends on time. Therefore, the "Hamiltonian" $\overline{H}$ governing the time evolution of the restricted wave function $\psi_{\bar{D}}(\mathbf{x}, t)$ must be a non-Hermitian operator. In this paper, we found an explicit representation of $\overline{H}$ and found that its non-Hermitian part can be written as a boundary term, which is proportional to the flux of the probability current operator through the boundary $\partial\bar{D}$ of $\bar{D}$. The explicit representation of $\overline{H}$ is given by Equation (25). From the time-dependent norm of $\psi_{\bar{D}}(\mathbf{x}, t)$, we computed the arrival time probability density, namely, the probability

that the particle will be detected as arriving at the detector between the times $t$ and $t + dt$, and found that this arrival probability density is equal to the flux of the probability current through the boundary $\partial \bar{D}$.

Our final result, i.e., that the arrival probability density is equal to the flux of the probability current, was also obtained by other approaches based on standard QM [5–8], and on Bohmian particle trajectories [9–12]. The approach of the present paper, which is also based on standard QM, is complementary to the existing approaches, because we arrived at the same conclusion using different methods. However, we stress that it is not generally accepted in the literature that the arrival probability density should be equal to the flux of the probability current (see [1–4] for reviews of other proposals). Thus, we believe that the result of this paper is a valuable contribution towards a resolution of an important problem in physics.

**Author Contributions:** Conceptualization, T.J. and H.N.; Formal analysis, T.J. and H.N.; Investigation, T.J. and H.N.; Methodology, T.J. and H.N.; Supervision, H.N.; Validation, T.J. and H.N.; Writing—original draft, T.J. and H.N.; Writing—review & editing, T.J. and H.N. All authors have read and agreed to the published version of the manuscript.

**Funding:** The work of T.J. was supported by Croatian Science Foundation Project No. IP-2020-02-9614.

**Data Availability Statement:** Data are contained within the article.

**Conflicts of Interest:** The funders had no role in the design of the study; in the collection, analyses, or interpretation of data; in the writing of the manuscript; or in the decision to publish the results.

## Appendix A. The Adjoint $\overline{H}^\dagger$ and Non-Hermiticity of $\overline{H}$

To extract the non-Hermitian part of $\overline{H}$, it is sufficient to find its adjoint $\overline{H}^\dagger$. Notice that while obtaining (21), we used the naïve rule for calculating the adjoint of the product of operators, $(AB)^\dagger = B^\dagger A^\dagger$, which is valid only in finite dimensional spaces or for some particular set of wave functions. Here, we will show that the non-Hermitian part of $\overline{H}$ can be extracted directly from its adjoint in a straightforward and rigorous way once the proper definition of the adjoint is used.

We start with the definition of the adjoint [16,17]:

*The adjoint $A^\dagger : \mathcal{D}(A^\dagger) \longrightarrow \mathcal{H}$ (with $\mathcal{D}$ denoting the domain of the operator) of a densely defined linear operator $A : \mathcal{D}(A) \longrightarrow \mathcal{H}$ is defined by*

- $\mathcal{D}(A^\dagger) := \{\psi \in \mathcal{H} \mid \exists \eta \in \mathcal{H} : \forall \alpha \in \mathcal{D}(A) : \langle \psi | A\alpha \rangle = \langle \eta | \alpha \rangle\}$;
- $A^\dagger \psi = \eta$.

Then, we analyze the expression in (13) and notice the following chain of equalities:

$$
\begin{aligned}
\langle \psi | \overline{H} | \varphi \rangle &\equiv \int_{\bar{D}} d^3x \, \psi^*(\mathbf{x}) \left[ -\frac{\boldsymbol{\nabla}^2}{2m} + V(\mathbf{x}) \right] \varphi(\mathbf{x}) \\
&= \int_{\mathbb{R}^3} d^3x \, \psi^*(\mathbf{x}) \chi_{\bar{D}}(\mathbf{x}) \hat{H} \varphi(\mathbf{x}) \\
&= \langle \psi | \bar{\pi} H | \varphi \rangle,
\end{aligned}
\tag{A1}
$$

where $\chi_{\bar{D}}(\mathbf{x})$ is the characteristic function of the region $\bar{D}$

$$
\chi_{\bar{D}}(\mathbf{x}) = \begin{cases} 1 & \text{for } \mathbf{x} \in \bar{D} \\ 0 & \text{for } \mathbf{x} \notin \bar{D}, \end{cases}
\tag{A2}
$$

and $\bar{\pi}$ is defined in (7). This means that

$$
\langle \mathbf{x} | \overline{H} | \varphi \rangle = \langle \mathbf{x} | \bar{\pi} H | \varphi \rangle = \chi_{\bar{D}}(\mathbf{x}) \hat{H} \varphi(\mathbf{x}),
\tag{A3}
$$

where $\varphi \in \mathcal{D}(H)$ and $\psi \in \mathcal{H}$. Now, in order to obtain the adjoint $\overline{H}^{\dagger}$, we need to derive $\eta$ from the general definition for the adjoint. This will enable us to find both the domain and the "rule of action" of the operator $\overline{H}^{\dagger}$. For that matter, we will use the simple identity

$$\psi^* \boldsymbol{\nabla}^2 \varphi = \boldsymbol{\nabla}[\psi^* \boldsymbol{\nabla} \varphi - (\boldsymbol{\nabla} \psi^*)\varphi] + (\boldsymbol{\nabla}^2 \psi^*)\varphi \tag{A4}$$

in order to rewrite (A1) as

$$\begin{aligned}
\langle \psi | \overline{H} | \varphi \rangle = \int_{\bar{D}} d^3 x \left( -\frac{\boldsymbol{\nabla}^2 \psi^*(\mathbf{x})}{2m} + V(\mathbf{x})\psi^*(\mathbf{x}) \right) \varphi(\mathbf{x}) \\
- \frac{1}{2m} \int_{\bar{D}} d^3 x \, \boldsymbol{\nabla}[\psi^* \boldsymbol{\nabla} \varphi - (\boldsymbol{\nabla} \psi^*)\varphi].
\end{aligned} \tag{A5}$$

The first term is equal to

$$\begin{aligned}
\int_{\bar{D}} d^3 x \left( -\frac{\boldsymbol{\nabla}^2 \psi^*(\mathbf{x})}{2m} + V(\mathbf{x})\psi^*(\mathbf{x}) \right) \varphi(\mathbf{x}) \\
= \int_{\mathbb{R}^3} d^3 x \left( \chi_{\bar{D}} \hat{H} \psi \right)^* \varphi \\
= \langle \overline{H} \psi | \varphi \rangle,
\end{aligned} \tag{A6}$$

while, for the second term, we have

$$\begin{aligned}
- \frac{1}{2m} \int_{\bar{D}} d^3 x \, \boldsymbol{\nabla}[\psi^* \boldsymbol{\nabla} \varphi - (\boldsymbol{\nabla} \psi^*)\varphi] \\
= -\frac{1}{2m} \int_{\partial \bar{D}} d\mathbf{S} \cdot [\psi^* \boldsymbol{\nabla} \varphi - (\boldsymbol{\nabla} \psi^*)\varphi] \\
= \frac{i}{2m} \int_{\partial \bar{D}} d\mathbf{S} \cdot (\langle \psi | \mathbf{x} \rangle \langle \mathbf{x} | \mathbf{p} \varphi \rangle + \langle \psi | \mathbf{p} | \mathbf{x} \rangle \langle \mathbf{x} | \varphi \rangle) \\
= \langle \psi | \frac{i}{2m} \int_{\partial \bar{D}} d\mathbf{S} \cdot \{ |\mathbf{x}\rangle \langle \mathbf{x}|, \mathbf{p} \} | \varphi \rangle \\
= \langle i \int_{\partial \bar{D}} d\mathbf{S} \cdot \mathbf{J} \psi | \varphi \rangle,
\end{aligned} \tag{A7}$$

where we use the Gauss theorem and the definitions (17) and (24), the $\partial \bar{D}$ is the boundary of $\bar{D}$, and $d\mathbf{S}$ is the area element directed outwards from $\bar{D}$. Here, we notice that the identity (A4) is only valid if $\psi$ is at least a function of the class $\mathcal{C}^2(\mathbb{R}^3)$ and, therefore, Equation (A5) is well defined only if $\psi \in \mathcal{H}$ is such that $\boldsymbol{\nabla}^2 \psi \in \mathcal{H}$, which will define the domain of the adjoint $\mathcal{D}(\overline{H}^{\dagger})$. Equations (A5)–(A7) together lead to

$$\begin{aligned}
\langle \psi | \overline{H} \varphi \rangle = \langle \left( \overline{H} + i \int_{\partial \bar{D}} d\mathbf{S} \cdot \mathbf{J} \right) \psi | \varphi \rangle \\
= \langle \overline{H}^{\dagger} \psi | \varphi \rangle
\end{aligned} \tag{A8}$$

and we can explicitly read out the "rule of acting" for the adjoint operator $\overline{H}^{\dagger}$

$$\overline{H}^{\dagger} = \overline{H} + i \int_{\partial \bar{D}} d\mathbf{S} \cdot \mathbf{J}, \tag{A9}$$

which, together with the domain $\mathcal{D}(\overline{H}^{\dagger})$, fully defines the operator $\overline{H}^{\dagger}$. Equation (A9) explicitly shows the non-Hermiticity of $\overline{H}$ and is in complete agreement with Equation (23).

The non-Hermiticity of $\overline{H}$ can also be described by the operator

$$N \equiv i(\overline{H} - \overline{H}^{\dagger}) = \int_{\partial \bar{D}} d\mathbf{S} \cdot \mathbf{J}. \tag{A10}$$

This operator is related to the arrival time distribution (36) via its expectation value

$$\mathcal{P}(t) = \langle \psi(t) | N | \psi(t) \rangle = \int_{\partial \bar{D}} d\mathbf{S} \cdot \mathbf{j}(\mathbf{x}, t), \tag{A11}$$

which can also be written as a volume integral

$$\begin{aligned}
\mathcal{P}(t) &= \int_{\bar{D}} d^3x \, \boldsymbol{\nabla} \cdot \mathbf{j}(\mathbf{x}, t) \\
&= \int_{\mathbb{R}^3} d^3x \, \chi_{\bar{D}}(\mathbf{x}) \boldsymbol{\nabla} \cdot \mathbf{j}(\mathbf{x}, t) \\
&= -\int_{\mathbb{R}^3} d^3x (\boldsymbol{\nabla} \chi_{\bar{D}}(\mathbf{x})) \cdot \mathbf{j}(\mathbf{x}, t).
\end{aligned} \tag{A12}$$

## Note

[1]    Notice that $V(t) = e^{-i\overline{H}t}$ is a contraction operator that forms a strongly continuous semi-group on the projected Hilbert space $\bar{\pi}\mathcal{H}$ and satisfies [8]

$$\frac{dV(t)}{dt} = -i\overline{H}V(t).$$

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
