# Peer review of "Arrival Time from Hamiltonian with Non-Hermitian Boundary Term"

_universe, doi:10.3390/universe10010035_

Round 1

Reviewer 1 Report

Comments and Suggestions for Authors

The manuscript addresses the arrival time problem in quantum mechanics, which involves predicting the probability distribution of the time a quantum particle will be detected at a specific region. The approach in this paper is motivated by the goal of avoiding mathematical subtleties that appeared in previous works.

The paper's findings provide additional theoretical evidence for the relationship between arrival probability density and probability current flux. After a small English editing of the manuscript, I recommend its publication.

Comments on the Quality of English Language

The article's English is easy to understand, but it requires editing. There are several instances where commas have been left out, and some words need to be replaced. Additionally, the word "so" has been used repeatedly throughout the article. To enhance the article's formality, I would recommend replacing "so" with more formal words such as "thus" or "therefore."

Author Response

Response to Reviewer 1:

According to the suggestion by the reviewer, we have corrected some typos, added some commas, and replaced some "so" -s with more formal expressions. Since those changes are minor, we do not indicate the exact places where such changes are made. 

Reviewer 2 Report

Comments and Suggestions for Authors

Referee Report on "Arrival Time from Hamiltonian with Non-Hermitian Boundary Term"

Title: "Arrival Time from Hamiltonian with Non-Hermitian Boundary Term"

Summary:
The paper presents a novel approach to determining quantum arrival time using a Hamiltonian with a non-Hermitian boundary term. The key focus is on the mathematical formulation of this Hamiltonian and its implications for calculating the quantum probability density of arrival at a detector. The approach involves integrating a boundary term into the Hamiltonian, which is proportional to the flux of the probability current operator through the boundary, leading to the conclusion that the arrival probability density is equivalent to this flux.

Referee Report:
1. **Mathematical Formulation**: The derivation of the non-Hermitian Hamiltonian, specifically in Equations 25 and 36, is a significant contribution. However, a clearer explanation of these equations and their practical implications would improve the paper.

2. **Treatment of Wavefunction Collapse**: The omission of a discussion on wavefunction collapse post-detection is a notable gap. Addressing this, even briefly, would provide a more comprehensive understanding of the model in the context of quantum measurement.

3. **Clarity in Specific Sections**: Sections discussing the theoretical implications of the non-Hermitian Hamiltonian could be more detailed. Specifically, a deeper analysis in Section 3 would be beneficial, where the connection between the mathematical model and physical interpretation could be more explicitly drawn.

4. **Experimental Implications**: Including potential experimental applications or tests of the proposed model would strengthen the paper, bridging the gap between theory and practice.

5. **Broader Context**: The paper would benefit from a discussion on how this model compares with existing approaches to the quantum arrival time problem, providing a clearer picture of its place within the broader quantum mechanics framework.

Overall, the paper makes an interesting contribution to quantum mechanics, especially in the study of quantum arrival time. Enhancements in the clarity of complex sections, inclusion of wavefunction collapse, and discussion of experimental applications would significantly elevate its impact.

Comments on the Quality of English Language

The use of english is ok.

Author Response

Response to Reviewer 2:

Reviewer's comment:
1. **Mathematical Formulation**: The derivation of the non-Hermitian Hamiltonian, specifically in Equations 25 and 36, is a significant contribution. However, a clearer explanation of these equations and their practical implications would improve the paper.

Authors's reply:
After Eq. (25) we add a somewhat long new paragraph beginning with 
"Eq. (25) is the main new result of this paper, so let us discuss its significance qualitatively. ..."
After Eq. (36) we add a new paragraph reading as follows:
"Mathematically, the final formula (36) is rather compact and general. The same formula has also been obtained in [8] by different methods, while here we derived it through the use of the restricted Hamiltonian (25) with the anti-hermitian boundary term."

----

Reviewer's comment:
2. **Treatment of Wavefunction Collapse**: The omission of a discussion on wavefunction collapse post-detection is a notable gap. Addressing this, even briefly, would provide a more comprehensive understanding of the model in the context of quantum measurement.

Authors's reply:
After Eq. (10) we add a new paragraph reading as follows:
"The restricted wave function can also be interpreted in terms of wave function collapse. When a part of the wave function enters the detector region D, there is a non-zero probability that the detector will detect the particle, i.e., that the wave function will collapse to the region D. But there is also a probability that the detector will not detect the particle, in which case we know that the particle is still outside of the detector region D, so the wave function collapses to the region DÌ„. Thus the restricted wave function can be interpreted as the collapsed wave function, corresponding to a negative outcome of measurement by the
detector. For more details of such an interpretation see [8]."

----

Reviewer's comment:
3. **Clarity in Specific Sections**: Sections discussing the theoretical implications of the non-Hermitian Hamiltonian could be more detailed. Specifically, a deeper analysis in Section 3 would be beneficial, where the connection between the mathematical model and physical interpretation could be more explicitly drawn.

Authors's reply:
At the end of Section 3 we add a new paragraph reading as follows:
"Physically, the most important consequence of evolution governed by the restricted Hamiltonian (25) with an anti-hermitian boundary term is the change of norm of the restricted wave function, as described by (30). The result (30) is rather intuitive, it can be visualized as a leak of wave function from the region DÌ„, where the flux of the probability current quantifies how much of the wave function leaks through the boundary of DÌ„."

----

Reviewer's comment:
4. **Experimental Implications**: Including potential experimental applications or tests of the proposed model would strengthen the paper, bridging the gap between theory and practice.

Authors's reply:
At the end of Section 4 we add a new paragraph reading as follows:
"The measurable predictions of the arrival time distribution based on flux of the probability current can in principle be distinguished experimentally from predictions of the arrival time distribution based on other approaches. It is beyond the scope of the present paper to discuss such measurable differences in detail, but they have been studied elsewhere [4]."

----

Reviewer's comment:
5. **Broader Context**: The paper would benefit from a discussion on how this model compares with existing approaches to the quantum arrival time problem, providing a clearer picture of its place within the broader quantum mechanics framework.

Authors's reply:
In Introduction we add two new paragraphs, beginning with
"In general, quantum mechanics makes unambiguous probabilistic predictions ..."
and
"One class of possibilities (see e.g. [1, 4] and references therein) is to reformulate quantum mechanics ..."

Reviewer 3 Report

Comments and Suggestions for Authors

The paper is devoted to another obtaining the result on the arrival time distribution as given by the flux of the probability correct. This result has been derived before. It may be published after discussion of a couple of items. First, the physical problem should be explicitly stated, formulated and discussed its physical applicability. The existence of Ref. is not enough. Second, detection region D should be described and discussed. Evidently, it is a 3-dimensional one. Which region of it correspond to time detection?

Author Response

Response to Reviewer 3:

Reviewer's comment:
First, the physical problem should be explicitly stated, formulated and discussed its physical applicability. The existence of Ref. is not enough.

Authors's reply:
In Introduction we add two new paragraphs, beginning with
"In general, quantum mechanics makes unambiguous probabilistic predictions ..."
and
"One class of possibilities (see e.g. [1, 4] and references therein) is to reformulate quantum mechanics ..."

----

Reviewer's comment:
Second, detection region D should be described and discussed. Evidently, it is a 3-dimensional one. Which region of it correspond to time detection?

Authors's reply:
At the end of Section 2 we add a new paragraph reading as follows:
"From a theoretical point of view, the exact specification of the detector region D is in our analysis still somewhat ambiguous. In principle, one could specify it by a more detailed model. But in practice, we believe that an experimental approach would be more fruitful. One could take an actual detector and impinge on it particles with wave functions which are very narrow in the position space, so that particles are effectively “classical” in the sense that their arrival time can be predicted as a classical deterministic event. In this way, one can determine the relevant detector region D experimentally, for the specific detector
at hand. After that, once D is known, one can make non-trivial theoretical analysis with “truly quantum” non-narrow wave functions."